# Post-Viral Fatigue Following SARS-CoV-2 Infection during Pregnancy: A Longitudinal Comparative Study

**DOI:** 10.3390/ijerph192315735

**Published:** 2022-11-26

**Authors:** Ana Maria da Silva Sousa Oliveira, Mariana Azevedo Carvalho, Luis Nacul, Fábio Roberto Cabar, Amanda Wictky Fabri, Stela Verzinhasse Peres, Tatiana Assuncao Zaccara, Shennae O’Boyle, Neal Alexander, Nilton Hideto Takiuti, Philippe Mayaud, Maria de Lourdes Brizot, Rossana Pulcineli Vieira Francisco

**Affiliations:** 1Disciplina de Obstetrícia, Departamento de Obstetrícia e Ginecologia da Faculdade de Medicina da Universidade de São Paulo, São Paulo 05403-000, Brazil; 2Complex Chronic Diseases Program—BC Women’s Hospital and Health Centre, Vancouver, BC V6H 3N1, Canada; 3Faculty of Infectious & Tropical Diseases, London School of Hygiene & Tropical Medicine, London WC1E 7HT, UK; 4UK Health Security Agency, London SW1P 3JR, UK; 5Faculty of Epidemiology & Population Health, London School of Hygiene & Tropical Medicine, London WC1E 7HT, UK; 6Divisão de Obstetrícia e Ginecología, Hospital Universitário da Universidade de São Paulo, São Paulo 05508-000, Brazil

**Keywords:** SARS CoV-2, COVID-19, post-viral fatigue, pregnancy, post-acute sequelae of SARS-CoV-2 infection (PASC)

## Abstract

Studies reported post-COVID-19 fatigue in the general population, but not among pregnant women. Our objectives were to determine prevalence, duration, and risk factors of post-viral fatigue among pregnant women with SARS-CoV-2. This study involved 588 pregnant women with SARS-CoV-2 during pregnancy or delivery in Brazil. Three groups were investigated: G1 (*n* = 259, symptomatic infection during pregnancy); G2 (*n* = 131, positive serology at delivery); G3 (*n* = 198, negative serology at delivery). We applied questionnaires investigating fatigue at determined timepoints after infection for G1, and after delivery for all groups; fatigue prevalence was then determined. Cox regression was used to estimate hazard ratio (HR) and 95% CI of the risk of remaining with fatigue in G1. Overall fatigue prevalence in G1 at six weeks, three months and six months were 40.6%, 33.6%, and 27.8%, respectively. Cumulative risk of remaining with fatigue increased over time, with HR of 1.69 (95% CI: 0.89–3.20) and 2.43 (95% CI: 1.49–3.95) for women with moderate and severe symptoms, respectively. Multivariate analysis showed cough and myalgia as independent risk factors in G1. Fatigue prevalence was significantly higher in G1 compared to G2 and G3. Post-viral fatigue prevalence is higher in women infected during pregnancy; fatigue’s risk and duration increased with the severity of infection.

## 1. Introduction

The severe acute respiratory syndrome coronavirus type-2 (SARS-CoV-2) emerged in Eastern Asia in late 2019, causing the worldwide pandemic of COVID-19 (coronavirus-associated disease 2019) [1]. When symptomatic, the disease’s initial manifestations were flu-like symptoms, such as fever, cough, fatigue, and shortness of breath. Typically, affected individuals have displayed a variable extent of dyspnea, hypoxia and radiological signs [2]. Patients who have recovered from COVID-19 can develop a post-viral syndrome called “post-COVID-19 Syndrome”, which presents with profound and persistent fatigue, intolerance to exertion and post-exertional malaise, disturbed sleep/wake cycle, and neuro-cognitive symptoms [3]. 

Recent studies have shown that, following COVID-19, patients can present persistent physical symptoms, such as fatigue (15 to 87%), dyspnea (10 to 71%), chest pain or tightness (12 to 44%) and cough (17 to 34%), which lead to a reported worsened quality of life in 44.1% of the cases [4,5]. Although fatigue appears to resolve spontaneously for most patients, it can be profound and may last for three months or longer, especially among intensive care unit survivors [6,7].

Fatigue can be defined as an overwhelming and sustained feeling of exhaustion, a diminished ability to perform physical and mental work at the usual level and feelings of tiredness to exhaustion, creating a general condition of lack of relief which interferes with the individual’s ability to perform everyday activities [8].

The fatigue symptoms following COVID-19 may be multi-factorial or be caused by any prolonged illness, organ dysfunction or psychological factors triggered or enhanced by the pandemic’s vast consequences or the fears of getting COVID-19 to those affected [9,10,11]. However, prolonged and chronic fatigue following infectious diseases, particularly acute viral diseases, have been commonly described, and in some cases may lead to myalgic encephalomyelitis or chronic fatigue syndrome (ME/CFS) [12,13]. Fatigue is frequently reported as a COVID-19 symptom per se and may present after recovery from acute infection [14]. However, fatigue is also a common complaint in the general population, with 21 to 33% of patients seeking attention in primary care settings describing it as a critical symptom [15,16], and is more frequently reported by women [17].

While several studies have described post-COVID-19 fatigue in the general population, we are unaware of studies reporting post-COVID-19 fatigue experienced by women during or after a pregnancy. This population group could be assumed to experience more frequent effects of fatigue. In our analysis, we sought to determine the prevalence over time and factors associated with fatigue following a diagnosis of COVID-19 (symptomatic SARS-CoV-2 infection) during pregnancy, and positive serology at delivery (asymptomatic SARS-CoV-2 infection) compared to women with SARS-CoV-2 negative serology in Sao Paulo, Brazil. 

## 2. Materials and Methods

We conducted a longitudinal comparative study within a cohort of pregnant women attending antenatal care, maternity and postnatal clinics or the delivery unit of one of the two public hospitals belonging to the University of Sao Paulo (USP) medical complex, specifically the Hospital das Clinicas and the Hospital Universitario, between 1 May 2020, and 31 May 2021. 

Study participants included two groups of women identified with the SARS-CoV-2 infection: (i) women who had been diagnosed with COVID-19 (symptomatic SARS-CoV-2) during pregnancy (G1; *n* = 259); (ii) women with no investigation for SARS-CoV-2 infection during pregnancy and a positive SARS-CoV-2 serology at delivery (G2; *n* = 131); and (iii) a comparison group of pregnant women (G3; *n* = 198) with no investigation for SARS-CoV-2 during pregnancy and negative SARS-CoV-2 serology at delivery. G3 women were enrolled consecutively, following the identification of G2 women, from the same delivery unit and within the same day of delivery if they consented. During the study period, SARS-CoV-2 serology was offered systematically to all women delivering at the USP delivery suites. Pregnant women vaccinated for SARS-CoV-2 were not included. Figure 1 shows each group, its size and times of the questionnaire application.

The study was conducted according to the guidelines of the Declaration of Helsinki and approved by the Ethics Committee of Hospital das Clinicas—FMUSP, Sao Paulo, Brazil, and the Ethics Committee of Hospital Universitario, Sao Paulo, Brazil (CAAE: 30270820.3.0000.0068). The protocol was registered at ClinicalTrials.gov on 01 December 2020 (NCT04647994). All participants provided signed informed consent.

Questionnaires for fatigue evaluation were applied face-to-face by trained research assistants following a specific schedule for each group: G1 participants were interviewed at six weeks, three months and six months after their COVID-19 diagnosis, then again at delivery, and at six weeks, three months and six months after delivery, whichever follow-up visit came first. Participants from G2 and G3 were interviewed at delivery and six weeks, three months and six months after delivery. 

Four types of custom-built questionnaires were used to ascertain fatigue and related symptoms: (i) A screening questionnaire was applied at recruitment and at each follow-up visit, if the previous screening was negative for ‘significant’ fatigue. The questionnaire determined the presence or absence of fatigue, including the duration and any loss of activity, measured mental and physical fatigue, and pain on a 10-point scale. (ii) A follow-up fatigue screening questionnaire, with similar structure to the first one but with modifications to account for any changes in fatigue since last assessment, was applied at follow-up visits. (iii) An additional in-depth complementary fatigue questionnaire was administered to women screened positive for ‘significant fatigue’ at any visit using (i) or (ii), to identify presence and severity of symptoms associated with fatigue, possible comorbidities or other possible causes of fatigue. (iv) A follow-up for the additional ‘significant fatigue’ questionnaire at subsequent visits, which however omitted questions about comorbidities or other possible causes of fatigue already probed [18,19,20]. ‘Fatigue’ was identified when the pregnant women reported persistent or recurrent profound tiredness, weariness or fatigue. ‘Fatigue most of the time’ was considered when the patient indicated feeling fatigued more than 50% of the time and when they needed to substantially reduce some activities or could no longer do some routine activities. ‘Significant fatigue’ included the same requirements for ‘fatigue most of the time’ with the addition of a positive answer to the statement “the patient could do half or less than half, as much as she could”; the fatigue would not be due to a long-standing disease or illness, and fatigue would not disappear with rest. The questionnaire also included self-scoring items (from 0 if absent to 10 for worst) relating to how the participant felt over the previous seven days regarding fatigue, physical fatigue, mental fatigue and pain. The questionnaires were created in Portuguese for use with Brazilian populations, based on questionnaires (in English) that have been used by the UK ME/CFS Biobank for many years [18,21]. The instruments used in our study are available in Portuguese as Appendix A.

G3 participants were serologically tested at each visit to confirm that they had remained seronegative. Participants’ demographics, medical and obstetrical history were obtained directly from their antenatal and postnatal clinic notes or obstetrical charts. Data regarding SARS-CoV-2 investigations or COVID-19 diagnosis during pregnancy were obtained from the REDCap database developed explicitly for a cohort study of the incidence of SARS-CoV-2/COVID-19 during pregnancy developed by the research team at the same maternal units [18]. 

The SARS-CoV-2 diagnosis was based on serological and molecular assays performed at our hospital during the cohort mentioned above [22]. Briefly, pregnant women with suspected COVID-19 symptoms had samples from the nasopharynx and/or back of throat tested by real-time polymerase chain reaction (RT-PCR, RealStar SARS-CoV-2 RT-PCR kit 1.0 RUO, Altona Diagnostics, Hamburg, Germany) using the LightCycler 96 Instrument (Roche, Mannheim, Germany). In addition, a serological assay was systematically performed for all women presenting at the USP delivery units using the Roche Elecsys anti-SARS-CoV-2 E2G300 assay (Roche Diagnostics, Mannheim, Germany), a chemiluminescence assay detecting IgG antibodies against SARS-CoV-2 nucleocapsid protein. 

Participants with identified COVID-19 by molecular and/or serological methods (G1) were further classified into three sub-groups according to the severity of their acute infection: (1) women with mild symptoms not needing hospital admission; (2) women with moderate symptoms (oxygen saturation rate SpO_2_ < 94% and/or respiratory rate > 24 breaths per min) who needed hospital admission for supplemental catheter-administered oxygen; and (3) women with severe COVID-19 who needed hospital admission in intensive care units for mechanical ventilation, or because of multi-organ involvement.

Categorical variables were presented as absolute frequencies and percentages, and continuous variables as medians with interquartile range (IQR). The Chi-squared and Fisher’s exact tests were applied to assess the association between categorical variables. The Kruskal-Wallis test was used to test for differences between groups for quantitative variables.

Cox proportional risk model for Group 1 was applied to estimate the risk of remaining with fatigue (HR) and its 95% confidence intervals (CI). The outcome variable was made using the time from the first day with fatigue to the date of the last event without fatigue or loss of follow-up. For the multivariate model, independent covariates such as maternal age, number of comorbidities, pregnancy trimester and severity of COVID-19 during pregnancy were applied. We employed the method based on scaled Schoenfield residuals to test the proportional hazards assumptions. Differences were considered significant when the *p*-value was less than 0.05. 

The data were analyzed using Statistical Package for the Social Sciences (SPSS version 20, IBM, Armonk, NY, USA). STATA Intercooled version 15 was used for survival analysis.

## 3. Results

Over the 13-month study period (1 May 2020 to 31 May 2021), 1170 pregnant women were recruited to the main cohort of SARS-CoV-2 incidence. Of the 1170, 627 were eligible to participate in this sub-analysis investigating post-viral fatigue, 31 women declined participation, and 8 were excluded because they were asymptomatic positive SARS-CoV-2 during antenatal care, leaving 588 (94%) for inclusion in the analysis. Of these, 259 were women with COVID-19 (symptomatic SARS-CoV-2 infection) during antenatal care (G1), 131 had positive SARS-CoV-2 serology at delivery with no direct evidence of SARS-CoV-2 infection during pregnancy (G2), and 198 were the seronegative control groups for G2 women (G3). Detailed characteristics of the three groups are shown in Table 1. The median gestational age at SARS-CoV-2 diagnosis in G1 was 27.4 weeks (IQR, 21.9–32.9).

### Prevalence over Time of Post-Viral Fatigue

The overall number of fatigue-assessment visits per participant ranged from 1 to 7, and 30.1% (177/588) participants attended at least 3 visits. Appendix A shows the frequency of visits by study group. 

Table 2 displays the frequency of fatigue-related outcomes after COVID-19 was diagnosed during pregnancy (G1). The prevalence of fatigue at 6 weeks, 3 months and 6 months post-diagnosis was 40.6% (95% CI: 33.3–47.9), 33.1% (95% CI: 25.3–41.9) and 27.5% (95% CI: 19.9–35.7), respectively.

In symptomatic women infected during pregnancy (G1), the cumulative risk of remaining with fatigue increases over time according to the severity of the disease (Figure 2). The group of patients with mild disease had a longer follow-up time, possibly because the disease did not influence the delivery time. Cases with severe COVID-19 had a significantly higher risk (HR = 2.43, 95% CI: 1.49–3.95) of remaining fatigued compared with those with mild disease (Table 3). In addition, among G1 women, there was no difference in the duration of post-viral fatigue in relation to the presence and number of comorbidities or trimester of pregnancy (Table 3). There was a tendency for fatigue to last longer with greater age, but this was not statistically significant. In addition, none of the investigated maternal comorbidities independently influenced the risk of fatigue in the multivariate analysis, with the following HR for hypertension, cardiac disease, and lung disease being HR = 1.15 (95% CI: 0.72–1.85), HR = 0.89 (95% CI: 0.33–2.42), HR = 1.13 (95% CI: 0.65–1.95), respectively. Regarding diabetes mellitus and mental disease in G1, none of the patients with fatigue had these comorbidities.

In addition, symptoms at the time of infection (cough, myalgia, dyspnea, fever, fatigue, asthenia, headache, diarrhea, runny nose, anosmia, sore throat, and dysgeusia) were associated with the risk of remaining with fatigue in univariate analysis (Table 4). In the multivariable analysis, the following were retained due to statistical significance: cough (HR = 1.76; 95% CI: 1.07–2.96), myalgia (HR = 1.57; 95% CI: 1.01–2.44), and anosmia (HR = 0.60; 95% CI: 0.40–0.88). Anosmia was a protective factor in the risk of remaining with fatigue. 

Table 5 displays the frequencies and comparisons of fatigue-related outcomes in the three groups by follow-up visits from delivery onwards. At all four-time points, there were significant differences in the frequency of overall fatigue between the three groups (G1 vs. G2; G1 vs. G3). Prevalence of fatigue was significantly higher at delivery, six weeks, three months and six months after delivery in G1 compared to respective visits in G2 and G3. For example, at six months after delivery, fatigue was reported by nine (16.4%) women in G1 compared to none in G2 and one (0.3%) in G3 (*p* = 0.021). Regarding ‘fatigue most of the time’ and ‘significant fatigue’ outcomes, no statistical test was applied for comparisons due to the very low frequency of events in all groups (Table 5). In addition, no difference was observed in the fatigue and pain scores between the groups at these time points (Appendix A).

## 4. Discussion

This study investigated the prevalence and persistence of post-viral fatigue after a diagnosis of SARS-CoV-2 infection during pregnancy or delivery among nearly 600 women. There were several significant findings. First, the prevalence of overall fatigue following COVID-19 was significantly higher compared to groups with positive serology at delivery but no evidence of infection during pregnancy (G2) or negative serology at delivery and in subsequent follow-ups. Second, approximately a quarter of pregnant women with COVID-19 during pregnancy presented fatigue six months after infection. Third, the risk of fatigue increased with the severity of the acute infection during pregnancy, with a 2.4-fold increased risk in women with severe disease compared to mild disease. On the other hand, there was no increase in the risk of presenting fatigue according to the trimester of infection or background comorbidity. Fourth, classic COVID-19 symptoms, such as cough and myalgia, were significantly associated with the risk of fatigue. Conversely, anosmia was a protective factor for fatigue. 

The observed occurrence of overall fatigue in our population six weeks, three months, and six months after initial diagnosis of COVID-19 during antenatal care were 40.6%, 33.6%, and 27.8%, respectively. Other studies have reported an even higher occurrence of post-viral fatigue following a COVID-19 diagnosis in different populations than pregnant women. However, rates vary between studies. Goertz et al. in the Netherlands described extremely high rates of fatigue (87%), 79 days after infection in the general population (85.3% of nonpregnant women), although that study may have suffered from selection and reporting biases as only patients with persistent COVID-19 symptoms were recruited through an internet survey and an online self-administered questionnaire [5]. Another publication from Mexico [23] reported a 53% (69/119) fatigue prevalence among patients three months after discharge from the hospital, of whom about 40% were women. However, we did not find any previous reports investigating post-viral fatigue after a diagnosis of SARS-CoV-2 infection in the specific group of pregnant women to compare with our data.

Importantly, our study also measured various standard outcomes of fatigue, with overall fatigue, fatigue most of the time, and significant fatigue in the three different groups after delivery (Table 5). In G1, 6 months after delivery, 16.4% of women reported overall fatigue, 1.8% of women with fatigue most of the time, but none reported significant fatigue. Overall, ‘significant fatigue’ was an uncommon event in our cohort (4 of 588 individuals), a finding that differs from other studies reporting higher rates in the context of post-COVID-19 [6,24]. This could be explained, as most of the studies did not differentiate between significant fatigue and overall fatigue, and by the fact that our population was an unselected cohort at various stages of SARS-CoV-2 infection severity. Moreover, the rates of ‘fatigue most of the time’ and ‘significant fatigue’ were similar to the prevalence of chronic diseases-associated ME/CFS, varying between 0.1% and 0.7% (median of 0.4%) [25].

On the other hand, the frequency of the outcome “fatigue” as a symptom is more directly comparable with rates measured in studies investigating post-COVID-19 fatigue. A meta-analysis of post-COVID-19 fatigue based on 25,000 cases found a prevalence of 32% (95% CI: 27–37) at 12 weeks or more following diagnosis in the general population. However, most patients had been hospitalized, and children were also included [26]. Our results indicated that one-third (33.6%) of women still had fatigue three months after contracting SARS-CoV-2 infection during pregnancy. However, small numbers at follow-up and the potential selection of returnees may have biased these results.

The reasons for fatigue occurrence during or after SARS-CoV-2 infection or COVID-19 remain poorly elucidated. Until COVID-19, most studies had focused on ME/CFS and measuring fatigue incidence in chronic illnesses. Some mechanisms have been described, such as inflammation and dysregulation of the hypothalamic–pituitary–adrenal axis and the autonomic nervous system [27]. Fatigue following acute infection has been studied following many agents such as dengue, Chikungunya, Ebola virus, mononucleosis, Rock River fever, Lyme and others [28,29,30]. The mechanisms of post-infectious fatigue related to COVID-19 likely overlap with those originated following other infections, except for the operation of agent-specific factors (e.g., chronic lung disease following COVID-19), and seem to suggest primarily immuno-neurological mechanisms, leading to a range of downstream abnormalities, such as micro-circulatory dysregulation, and intracellular and extracellular mitochondrial dysfunction [31]. However, none of these have yet been implicated in long COVID-19 or SARS-CoV-2 post-viral fatigue [32]. 

In this study, we have investigated factors that may influence the risk of having post-viral fatigue when SARS-CoV-2 infection happens during pregnancy. The most significant factor was the severity of the initial disease, with patients admitted to ICU having the highest risk, while no risk was observed for infected women without symptoms. Other factors, such as maternal age and comorbidities, did not influence the risk of fatigue, although our sample size may have been limited to investigate these effects. 

This study had several strengths, including a relatively large number of COVID-19 cases recruited from a single center; the inclusion of comparative SARS-CoV-2 seropositive and seronegative groups; its longitudinal design with several follow-up visits at predefined time points to determine fatigue outcomes; the application of detailed and validated fatigue outcome measurement tools; and the opportunity to estimate risks according to the severity of COVID-19. This publication originality derives from its population of pregnant women, a group that may conceivably be at higher risk of fatigue. However, as fatigue is not an unusual symptom in pregnancy and post-partum, many women may not have reported it, leading to some underestimates.

We are also aware of our study limitations. Firstly, we had a low visit completion rate, which can be explained by the low social-economic status of the group of women coming to the University of Sao Paulo maternity (i.e., users of the Brazilian public health system). These women may have incurred significant difficulties in using the public transportation system during the COVID-19 pandemic from their distant suburban dwellings and may have perceived the potentially high risk of travel with little incentive or benefit from attendance. This low attendance rate might have led to an overestimation of risk, if originally sicker patients were more inclined to attend, or an underestimation of risk, if they had decided to stay home plagued with fatigue or for other reasons, e.g., to avoid further exposure to infection. Fewer visits will have led to more limited power to find significant associations. A more convenient means of follow-up using phones and the internet could have been used, but this may have excluded participants with low access to modern communication technology. Secondly, we cannot entirely pinpoint when infection occurred in women only found seropositive at delivery; hence the estimation of incidence or duration of fatigue cannot be precise. Finally, these custom-built questionnaires were not validated in the Brazilian population, which results in potential bias in the ascertainment of outcomes. However, this possible bias was mitigated by the fact that the instruments were applied face-to-face by trained research assistants, minimizing the risk of patients misunderstanding the questions.

## 5. Conclusions

Acquisition of SARS-CoV-2 during pregnancy was associated with a higher prevalence of post-viral fatigue. Moreover, the risk and duration of fatigue increased with the severity of the infection.

## Figures and Tables

**Figure 1 ijerph-19-15735-f001:**
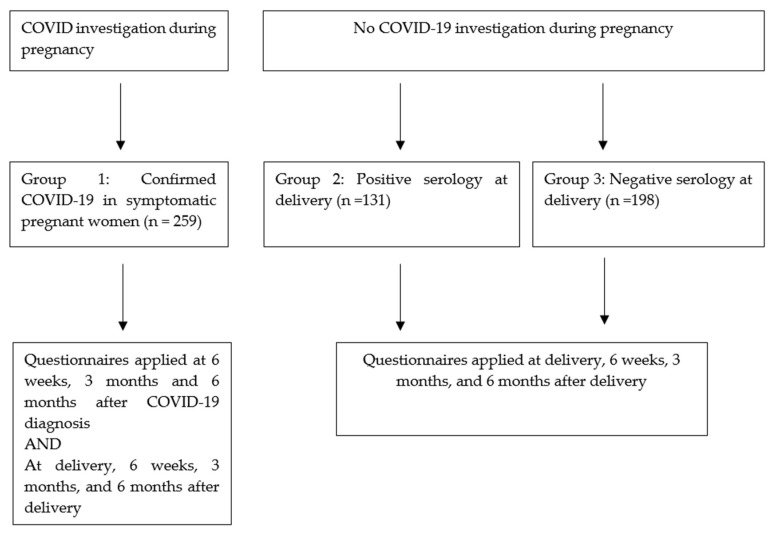
Details on each group, its size, and times of questionnaire application.

**Figure 2 ijerph-19-15735-f002:**
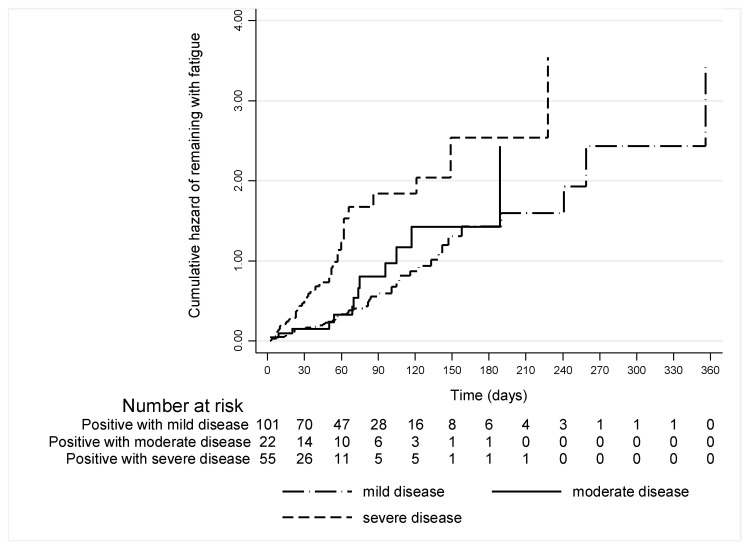
Cumulative risk of remaining with fatigue in women with COVID-19 identified during pregnancy (G1) according to the severity of disease (mild, moderate, and severe).

**Table 1 ijerph-19-15735-t001:** Population characteristics (*n* = 588).

Maternal Characteristics	*n* (%) or Median (IQR)		
	Group 1 (*n* = 259)	Group 2 (*n* = 131)	Group 3 (*n* = 198)
Age, years, median	31.0 (26–36)	27.0 (23–75)	30.5 (24–36)
Parity			
Nulliparous	78/256 (30.1)	59/131 (44.7)	75/198 (37.9)
Multiparous	178/256 (68.7)	73/131 (55.3)	123/198 (62.1)
Ethnicity			
White	124/257 (48.2)	44/131 (33.6)	74/196 (37.8)
Other	133/257 (51.8)	87/131 (66.4)	122/196 (62.2)
Current smoking	19/248 (7.7)	9/131 (6.9)	19/197 (9.6)
Comorbidities			
Diabetes mellitus	10/258 (3.9)	5/131 (3.8)	5/197 (2.5)
Hypertension	40/258 (15.5)	5/131 (3.8)	18/197 (9.1)
Heart disease	13/258 (5.0)	2/131 (1.5)	6/197 (3.0)
Lung disease	29/258 (11.2)	3/131 (2.3)	12/197 (6.1)
Mental health condition	24/257 (9.3)	6/131 (4.6)	19/197 (9.6)
Number of comorbidities			
0	158/258 (61.8)	110/131 (84.0)	142/197 (72.1)
1	80/258 (28.1)	17/131 (13.0)	44/197 (22.3)
≥2	20/258 (7.8)	4/131 (3.1)	12/197 (6.1)
Obstetrical outcomes			
Vaginal delivery	79/217 (36.4)	49/129 (37.5)	81/194 (41.7)
Cesarean delivery	138/217 (63.6)	79/129 (61.2)	107/194 (55.2)
Curettage	0/217	2/129 (1.5)	6/194 (3.1)

Group 1 = pregnant women with COVID-19 diagnosed during antenatal care; Group 2 = pregnant women with positive SARS-CoV-2 serology at delivery; Group 3 = pregnant women with negative SARS-CoV-2 serology at delivery.

**Table 2 ijerph-19-15735-t002:** Prevalence over time of fatigue after COVID-19 was diagnosed during pregnancy at each time point after infection (G1, *n* = 259).

Fatigue-Related Outcomes	Timepoint after COVID-19		
	6 Weeks (*n* = 175) *n* (%)	3 Months (*n* = 125) *n* (%)	6 Months (*n* = 79) *n* (%)
Fatigue	71 (40.6)	42 (33.6)	22 (27.8)
Fatigue most of the time ^#^	6 (3.4)	5 (3.9)	5 (6.3)
Significant fatigue ^##^	2 (1.1)	3 (2.4)	1(1.3)

^#^ Fatigue most of the time: fatigue more than 50% of the time. ^##^ Significant fatigue: fatigue most of the time in addition to “the patient could do half or less than half, as much as she could.” and the fatigue would not be due to a long-standing disease or illness and would not disappear with rest.

**Table 3 ijerph-19-15735-t003:** Risk of remaining with fatigue according to maternal COVID-19 severity. Data presented here are for G1 only, *n* = 259, of whom 178 had fatigue at any point and are included here.

Variables	Hazard Ratio (HR)	95% CI	*p*
		Lower–Upper	
SARS-CoV-2 infection			
Positive with mild disease	Reference		
Positive with moderate disease	1.69	0.89–3.20	0.107
Positive with severe disease	2.43	1.49–3.95	<0.001
Age, years			
≤25	Reference		
25 to 34	1.43	0.64–3.23	0.383
≥35	2.08	0.95–4.80	0.084
Pregnancy trimester			
1st	Reference		
2nd	1.09	0.31–3.79	0.890
3rd	0.96	0.29–3.18	0.943
Comorbidities *			
0	Reference		
1	0.83	0.51–1.35	0.460
≥2	0.67	0.28–1.60	0.372

* Comorbidities include any of the following: diabetes mellitus, hypertension, heart disease, lung disease, rheumatologic disease, and mental disease.

**Table 4 ijerph-19-15735-t004:** Univariate analysis of the risk of remaining with fatigue according to COVID-19 symptoms in G1 (*n* = 259).

Variables	Hazard Ratio (HR)	95% CI	*p*
		Lower–Upper	
Cough	1.80	1.08–2.99	0.024
Myalgia	1.51	0.98–2.33	0.060
Dyspnea	1.43	0.93–2.18	0.100
Fever	1.54	1.04–2.28	0.032
Fatigue	1.32	0.90–1.94	0.162
Asthenia	1.20	0.82–1.76	0.345
Headache	1.01	0.68–1.49	0.968
Diarrhea	1.33	0.82–2.15	0.245
Runny nose	0.72	0.49–1.06	0.092
Anosmia	0.66	0.45–0.97	0.033
Sore throat	0.76	0.49–1.18	0.218
Dysgeusia	0.67	0.46–0.99	0.042

**Table 5 ijerph-19-15735-t005:** Comparison of fatigue-related outcomes after delivery between the groups (*n* = 588).

Outcome at Each Time Point		Group 1 (*n* = 259) *n* (%)	Group 2 (*n* = 131) *n* (%)	Group 3 (*n* = 198) *n* (%)	*p*-Value
Fatigue	Delivery	28/145 (19.3)	1/125 (0.8)	4/181 (2.2)	<0.001 *
	Six weeks	34/134 (25.4)	2/56 (3.6)	3/69 (4.3)	<0.001 *
	Three months	18/78 (23.1)	3/34 (8.8)	2/44 (4.5)	0.011 **
	Six months	9/55 (16.4)	0/28	1/34 (2.9)	0.011 **
Fatigue most of the time ^#^	Delivery	2/145 (1.4)	0/125	2/181 (1.0)	NA
	6 weeks	2/134 (1.5)	1/56 (5.3)	0/69	NA
	3 months	2/78 (2.6)	0/34	0/44	NA
	6 months	1/55 (1.8)	NA	0/34	NA
Significant fatigue ^##^	Delivery	2/145 (1.4)	0/125	1/181 (0.6)	NA
	6 weeks	0/134	0/56	0/69	NA
	3 months	1/78 (1.3)	0/34	0/44	NA
	6 months	0/55	NA	0/34	NA

* Chi-squared test; ** Fisher exact test; ^#^ Fatigue most of the time: fatigue more than 50% of the time. NA = not applicable. ^##^ Significant fatigue: fatigue most of the time in addition to “the patient could do half or less than half, as much as she could” and the fatigue would not be due to a long-standing disease or illness and would not disappear with rest. Group 1= pregnant women with COVID-19 diagnosed during antenatal care; Group 2 = pregnant women with positive SARS-CoV-2 serology at delivery; Group 3 = pregnant women with negative SARS-CoV-2 serology at delivery.

## Data Availability

The datasets generated during and/or analyzed during the current study are available from the corresponding author upon reasonable request.

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
