# Peer review of "Post-Viral Fatigue Following SARS-CoV-2 Infection during Pregnancy: A Longitudinal Comparative Study"

_ijerph, 2022, doi:10.3390/ijerph192315735_

Round 1

Reviewer 1 Report

In the study by Oliveira et al, authors aimed to verify prevalence, duration, and risk factors of post-viral fatigue among pregnant women with SARS-CoV-2. I consider this study has a good sample size, and performed an adequate and reliable statistical analysis and data interpretation. Considering the strong relevance of the study and the other comments above, I suggest the publication of the manuscript with minor consideration.

Minor comments:

I suggest the authors provide a study timeline as figure 1. In the figure, the authors could show a scheme that summarize the groups and sample size, exclusions, times of questionnaire application, and times of follow-up. It would be helpful to the readers better understand your scheme of collecting the data.

Reviewer 2 Report

Well done to all authors ,i am speechless to all of your efforts. A few suggestions that you may consider;

1-In addition, symptoms at the time of infection (cough, myalgia, dyspnea, fever, fa- 213 tigue, asthenia, headache, diarrhea, runny nose, anosmia, sore throat, and dysgeusia) 214 were significantly associated with the risk of remaining with fatigue in univariate analysis 215 (Table 4)-----( were associated will be much more elegant)

2-In this study, this study (this words are used so many time in discussion section  if you may change the word ,discussion section will be more elite)

Author Response

Thank you very much for revising our work and for helping us improve our publication.

We have changed the wording in the manuscript to improve readability.

Reviewer 3 Report

The manuscript presented some results after the authors applied questionnaires investigating fatigue following SRAS-CoV_2 infection during pregnancy.

Some significant problems are regarded in these questionnaires. First, these questionnaires are not translated and validated for Brazilian patients. Second, the outcomes are unclear and the methods of measuring fatigue are not very clear.

Author Response

Thank you for helping us improve this manuscript.

We used questionnaires specially built for a study of post-viral fatigue syndrome following Zika infection (unpublished). The questionnaire was created in Portuguese for use with Brazilian populations, based on questionnaires (in English) that have been used by the UK ME/CFS Biobank for many years (references at https://pubmed.ncbi.nlm.nih.gov/28649428/and https://www.ncbi.nlm.nih.gov/pmc/articles/PMC6288193/).

They specifically take into account ME/CFS diagnostic criteria, with the questionnaires built based on the following guidelines:

  1. Myalgic Encephalomyelitis/Chronic Fatigue Syndrome: Clinical Working Case Definition, Diagnostic and Treatment Protocols (Bruce M. Carruthers, Anil Kumar Jain, Kenny L. De Meirleir, Daniel L. Peterson, Nancy G. Klimas, A. Martin Lerner, Alison C. Bested, Pierre Flor-Henry, Pradip Joshi, A. C. Peter Powles, Jeffrey A. Sherkey & Marjorie I. van de Sande (2003) Myalgic Encephalomyelitis/Chronic Fatigue Syndrome, Journal of Chronic Fatigue Syndrome, 11:1, 7-115, DOI: 10.1300/J092v11n01_02)
  2. Myalgic encephalomyelitis: International Consensus Criteria (Carruthers BM, van de Sande MI, De Meirleir KL, Klimas NG, Broderick G, Mitchell T, Staines D, Powles AC, Speight N, Vallings R, Bateman L, Baumgarten-Austrheim B, Bell DS, Carlo-Stella N, Chia J, Darragh A, Jo D, Lewis D, Light AR, Marshall-Gradisnik S, Mena I, Mikovits JA, Miwa K, Murovska M, Pall ML, Stevens S. Myalgic encephalomyelitis: International Consensus Criteria. J Intern Med. 2011 Oct;270(4):327-38. doi: 10.1111/j.1365-2796.2011.02428.x. Epub 2011 Aug 22. Erratum in: J Intern Med. 2017 Oct;282(4):353. PMID: 21777306; PMCID: PMC3427890.)
  3. The Chronic Fatigue Syndrome: A Comprehensive Approach to Its Definition and Study (Fukuda K, Straus SE, Hickie I, Sharpe MC, Dobbins JG, Komaroff A. The Chronic Fatigue Syndrome: A Comprehensive Approach to Its Definition and Study. Annals of Internal Medicine. 1994 Dec 15;121(12):953.)

We used these criteria sets as the basis because they are extensively used to assess fatigue and related symptoms in individuals with post-viral fatigue syndrome and ME/CFS due to other causes in different populations.

Our questionnaires were not formally validated in the Brazilian population, but we decided to use them because of the urgency of better understanding COVID-19 complications in pregnant and postpartum women. The instruments were applied face-to-face by trained research assistants, thereby reducing the bias of patients not understanding the questions. To improve clarity, we have now added the following sentences to the Methods section:

“The questionnaires were created in Portuguese for use with Brazilian populations, based on questionnaires (in English) that have been used by the UK ME/CFS Biobank for many years [21, 22]. The instruments used in our study are available in Portuguese as supplemental material”.

And the following to the Discussion section as a limitation:

“Finally, these custom-built questionnaires were not validated in the Brazilian population, which results in potential bias in the ascertainment of outcomes. However, this possible bias was mitigated by the fact that the instruments were applied face-to-face by trained research assistants, minimizing the risk of patients misunderstanding the questions”. 

Round 2

Reviewer 3 Report

The bias is a big disadvantage of your results and discussion. Every research based on a questionnaire must use a translated and validated questionnaire (visible in a published article) according to the international guidelines.